# OpenReview forum: "Don't Settle Too Early: Self-Reflective Remasking for Diffusion Language Models"
_ICLR.cc/2026/Conference — ICLR 2026 Poster_

### Official Review · Reviewer_AeNa · 2025-10-30

**Soundness:** 3
**Presentation:** 4
**Contribution:** 2
**Rating:** 4
**Confidence:** 4

**Summary:**

The paper proposes **RemeDi (Remasking-enabled Diffusion Language Model)**, which adds a self-reflective remasking mechanism to diffusion language models (DLMs). Traditional DLMs fix tokens once unmasked, making them unable to revise early mistakes. RemeDi introduces a dual-stream transformer: a **Token Prediction Stream (TPS)** predicts token distributions, while an **Unmasking Policy Stream (UPS)** predicts per-token confidence. Low-confidence tokens are re-masked and regenerated in later steps, allowing iterative self-correction. Training proceeds in two stages: **Remask SFT** (supervised fine-tuning) teaches the model to identify and remask incorrect tokens using mixed masking and random replacement; **Remask RL** fine-tunes full trajectories using task-specific rewards. Experiments on code, math, and reasoning benchmarks show that RemeDi surpasses prior DLMs (Dream, LLaDA, LLaDOU) and even matches or exceeds autoregressive baselines of similar size.

However, despite the promising empirical results, the paper exhibits several methodological and reporting weaknesses. Many experimental details are under-specified: the reward design for Remask RL is not clearly defined, and critical hyperparameters or normalization schemes are absent, even in the appendix. The paper does not report any compute or training-time comparisons, leaving unclear whether the observed gains stem from algorithmic innovation or simply greater computational cost. Similarly, convergence curves and efficiency analyses are missing, despite claims of faster training. The method section, while conceptually sound, omits low-level implementation details—such as how the UPS interacts with TPS during training or how remask thresholds are selected—making reproduction difficult. While the writing is clear at a high level, the technical exposition lacks sufficient depth and cross-references to equations and figures, resulting in incomplete methodological transparency.

**Strengths:**

1. **Addresses a concrete limitation of DLMs** — RemeDi directly tackles the inability to revise early decoding errors through a principled, learnable remasking mechanism.
2. **Clear and interpretable design** — The dual-stream architecture (TPS + UPS) separates token prediction from confidence estimation, enabling transparent remasking decisions.
3. **Comprehensive experiments** — Evaluations cover math (GSM8K, MATH), code (HumanEval, MBPP), and reasoning tasks (ARC-C, AlpacaEval), consistently improving over Dream and LLaDA.
4. **Empirical evidence for self-correction** — Qualitative examples and remask frequency analysis show that RemeDi learns to remask more frequently on harder or more structured tasks.
5. **Training stability and convergence** — Compared to LLaDOU RL, Remask RL converges faster and achieves higher accuracy under identical settings.

**Weaknesses:**

1. **Limited quantitative comparison to other edit-based diffusion models** (e.g., ReMDM, Seed Diffusion). The paper cites prior edit-based diffusion models (e.g., ReMDM, Seed Diffusion) but omits quantitative comparisons, leaving unclear how RemeDi’s improvements scale relative to these baselines.
2. **Under-specified reward formulation in Remask RL** — lacks details on normalization, weighting, or how rewards interact with confidence-based remasking.
3. **Compute and efficiency concerns** — The two-stage SFT → RL training likely increases overall cost, but compute parity with baselines is unreported, making it unclear whether improvements stem from algorithmic design or added compute.
4. **Figure placement and clarity** — Key visualizations (like the dynamic remasking behavior) appear in the appendix instead of the main paper.
5. **Missing convergence curves** — The claim that Remask RL converges as efficiently as other DLMs lacks quantitative support.
6. **Necessity of two-stage training unclear** — It is not shown whether a stronger SFT baseline or a joint training objective could achieve similar effects.
7. **Stability and variance during RL** — The paper lacks reporting on reward variance, remask rate dynamics, or regularization methods.
8. **Ablation under matched compute not provided** — It is unclear whether RL brings improvement beyond additional training time.

**Questions:**

1. **Comparison scope:** Why were models like *ReMDM* and *Seed Diffusion* not included in quantitative comparisons? Could a smaller-scale reimplementation or proxy be feasible?
2. **Reward design:** Please provide explicit formulas or examples of the reward functions used in the RL phase and clarify normalization or scaling strategies.
3. **Effectiveness of RL:** How does the RL-trained model differ behaviorally from the SFT-only version (e.g., fewer unnecessary remasks, faster correction)?
4. **Visualization and convergence:** Could you move the dynamic remasking visualization (currently in appendix) to the main paper and add training loss curves to verify efficiency claims?
5. **Compute accounting:** Please provide compute cost (GPU hours, wall time, token count) for both stages. How does SFT-only compare to SFT→RL at equal compute?
6. **Necessity of two-stage design:** Did you test a single-stage joint objective or stronger SFT baseline under matched compute?
7. **Stability:** What regularization methods (e.g., KL penalty, reward whitening, gradient clipping) were applied to stabilize RL?
8. **Matched compute ablation:** Could you run an SFT-only baseline using the same compute as SFT→RL to confirm that RL brings genuine benefit?
9. **Generalization to other DLMs:** Since the model is based on LLaDA-8B, could this remasking mechanism generalize to other diffusion LMs such as Dream? If so, what architectural adjustments are needed?
10. **Random alternative tokens:** In SFT, are random alternatives drawn uniformly from the vocabulary or sampled from the model’s top-k predictions?
11. **Inference efficiency:** The paper states RemeDi achieves faster convergence; could you include **inference latency comparisons** (e.g., tokens per second vs. LLaDA), given that TPS + UPS may add computational overhead?

---

> ### Author Response · Authors · 2025-12-03
>
> **W1: Limited quantitative comparison to other edit-based diffusion models (e.g., ReMDM, Seed Diffusion). The paper cites prior edit-based diffusion models (e.g., ReMDM, Seed Diffusion) but omits quantitative comparisons, leaving unclear how RemeDi’s improvements scale relative to these baselines.**
>
> Here we provide two separate sets of comparisons.
>
> **Training with Seed Diffusion.**
> Since the official Seed Diffusion model and implementation are not publicly available, we do our best to reproduce it. For a fair comparison, we train seed diffusion under the same base model and identical training configuration as in Appendix B.2.5, so that it can be directly compared with our Remask SFT. Our reproduced implementation is also provided together with our codebase (https://github.com/iiiutch-ii/RemeDi). As shown in Table I, Remask SFT consistently outperforms Seed Diffusion, demonstrating the advantage of learning an explicit remasking policy during training.
>
> **Table I: Unified head-to-head comparison with other training algorithms under identical settings.**
>
> | Method         | GSM8K    | MATH-500 | HumanEval | MBPP     |
> | -------------- | -------- | -------- | --------- | -------- |
> | Baseline       | 80.3     | 34.7     | 41.5      | 42.6     |
> | Vanilla SFT    | 83.1     | 40.1     | 48.2      | 43.4     |
> | Seed Diffusion | 63.9     | 28.0     | 5.4       | 9.8      |
> | Remask SFT     | **83.6** | **42.7** | **50.0**  | **44.0** |
>
> **Inference with ReMDM.** Separately, we apply the ReMDM predictor–corrector to RemeDi-Instruct using the same evaluation setup as in Appendix B.3.1. The results in Table II show that our learned remasking policy (via Remask SFT) is more effective than the random remasking strategy employed in ReMDM.
>
> **Table II: Comparison between our learned remask policy in RemeDi and the ReMDM predictor–corrector, both evaluated with RemeDi-Instruct.**
>
> | Method         | GSM8K    | MATH-500 | HumanEval | MBPP     |
> | -------------- | -------- | -------- | --------- | -------- |
> | RemeDi + ReMDM | 58.3     | 38.7     | 39.6      | 54.2     |
> | RemaDi (Ours) | **86.3** | **52.2** | **71.3**  | **57.8** |
>
> **W2 & Q2: Under-specified reward formulation in Remask RL — lacks details on normalization, weighting, or how rewards interact with confidence-based remasking. / Reward design: Please provide explicit formulas or examples of the reward functions used in the RL phase and clarify normalization or scaling strategies.**
>
> For a fair comparison, our RL training pipeline follows GRPO[1] (Group Relative Policy Optimization). We provide a full pseudocode in Algorithm 2 of the Appendix, and our training code in an anonymous repository (https://github.com/iiiutch-ii/RemeDi). GRPO computes advantage values using group-wise normalization: for each prompt, we generate a batch of roll-outs and compute the reward mean and standard deviation within that group. The normalized advantage $A=\frac{r-\text{mean}}{\text{std}}$ is then used to update the policy, including both the token-generation distribution and the confidence-based remasking policy (Alg. 2, L12–18). Thus, rewards directly shape when and how the model remasks tokens.
>
> Detailed reward definitions are provided in Appendix B.2.4. For verifiable rewards, a script is developed to validate if the generated response is correct(e.g., for code generation, we judge if it passes all test cases). A correct answer passing the validation yields $r=1$, otherwise $r=0$. For model-based rewards, a pre-trained reward model (RM) takes the prompt and model response as input and outputs a scalar score $r$ reflecting human preference. These rewards are naturally normalized by GRPO, no additional weighting or rescaling is applied.
>
>
> **Q7: Stability: What regularization methods (e.g., KL penalty, reward whitening, gradient clipping) were applied to stabilize RL?**
>
> For RL training stability, we only apply gradient clipping with a maximum norm of 1.0.

---

> > ### Author Response · Authors · 2025-12-03
> >
> > **Clarifying the Role of Remask RL (W6 & Q3 & Q6)**
> >
> > > W6: **Necessity of two-stage training unclear** — It is not shown whether a stronger SFT baseline or a joint training objective could achieve similar effects.
> > >
> > > Q3: **Effectiveness of RL:** How does the RL-trained model differ behaviorally from the SFT-only version (e.g., fewer unnecessary remasks, faster correction)?
> > >
> > > Q6: **Necessity of two-stage design:** Did you test a single-stage joint objective or stronger SFT baseline under matched compute?
> >
> > **W6 & Q6:** SFT → RL is a conventional training pipeline for both AR and diffusion LLMs, which is also conducted in many other works (e.g., LLaMA-2[2], LLaMA-3[3], DeepSeek[4]). We follow this for a fair comparison. In Sec. 4.3, we conduct ablation studies and demonstrate that under same training cost and configurations, Remask SFT and RL outperforms their conventional counterparts separately.
> >
> > **Q3:** To assess the behavioral effect of RL, we compare the per-token remask frequencies between the SFT-only model and the RL-trained model. Interestingly, the RL-trained model performs **more** remasking on average. As shown in Table III, higher remask frequency correlates with improved performance, suggesting that additional remasking provides more opportunities for RemeDi to detect and correct early-step errors.
> >
> > **Table III: Average remask frequency and performance on different tasks. Average remask frequency (ARF) denotes how many times each token is remasked on average during decoding.**
> >
> > | Model                 | GSM8K    | ARF on GSM8K | HumanEval | ARF on HumanEval | AlpacaEval | ARF on AlpacaEval |
> > | --------------------- | -------- | ------------ | --------- | ---------------- | ---------- | ----------------- |
> > | RemeDi (+ Remask SFT) | 86.3     | 0.16         | 71.3      | 0.070            | 12.5       | 0.012             |
> > | RemeDi (++ Remask RL) | **89.1** | **0.56**     | **73.2**  | **0.89**         | **24.8**   | **0.086**         |
> >
> > **SFT & RL Compute Cost and Matched-Compute Ablation (W3 & W8 & Q5 & Q8)**
> >
> > > W3: **Compute and efficiency concerns** — The two-stage SFT → RL training likely increases overall cost, but compute parity with baselines is unreported, making it unclear whether improvements stem from algorithmic design or added compute.
> > >
> > > W8: **Ablation under matched compute not provided** — It is unclear whether RL brings improvement beyond additional training time.
> > >
> > > Q5: **Compute accounting:** Please provide compute cost (GPU hours, wall time, token count) for both stages. How does SFT-only compare to SFT→RL at equal compute?
> > >
> > > Q8: **Matched compute ablation:** Could you run an SFT-only baseline using the same compute as SFT→RL to confirm that RL brings genuine benefit?
> >
> > **Q5:** We provide full compute accounting for both training stages:
> >
> > - **Remask SFT**: ~672 H800-days (3 weeks on 32xH800)
> > - **Remask RL**: ~32 H800-days (1 day on 32xH800)
> >   Total: **704 H800-days**.
> >
> > **W3 & W8 & Q8:** To directly address whether performance gains come from RL or merely additional training time, we conduct a **matched-compute ablation**: we continue training the RemeDi-Instruct model for an extra **2,000 steps** (which consumes approximately **32 H800-days**, the same compute as used in the RL stage) and evaluate its performance. The results are summarized below:
> >
> > **Table IV: Matched-compute ablation between extra SFT training and Remask RL.**
> >
> > |                              | GSM8K    | MATH-500 | HumanEval | MBPP     |
> > | ---------------------------- | -------- | -------- | --------- | -------- |
> > | RemeDi-Instruct              | 86.3     | 52.2     | 71.3      | 57.8     |
> > | + ~32 H800-days SFT training | 83.6     | 52.6     | 62.8      | 57.8     |
> > | + ~32 H800-days RL training  | **89.1** | **53.2** | **73.2**  | **59.4** |
> >
> > Further SFT provides marginal improvements, confirming the necessity of the Remask RL stage.

---

> > > ### Author Response · Authors · 2025-12-03
> > >
> > > **Paper Revision (W4 & W5 & W7 & Q4)**
> > >
> > > > W4: **Figure placement and clarity** — Key visualizations (like the dynamic remasking behavior) appear in the appendix instead of the main paper.
> > > >
> > > > W5: **Missing convergence curves** — The claim that Remask RL converges as efficiently as other DLMs lacks quantitative support.
> > > >
> > > > W7: **Stability and variance during RL** — The paper lacks reporting on reward variance, remask rate dynamics, or regularization methods.
> > > >
> > > > Q4: **Visualization and convergence:** Could you move the dynamic remasking visualization (currently in appendix) to the main paper and add training loss curves to verify efficiency claims?
> > >
> > > We appreciate the reviewer’s suggestions for improving clarity and transparency. In the revised manuscript, we will:
> > >
> > > 1. **W4 & Q4:** Due to the strict page limit, we are unable to move the dynamic remasking visualization from the appendix into the main paper. However, we will explicitly reference and describe the key insights from this figure in the main text.
> > > 2. **W5:** **Add reward curves** in Appendix C, Fig. 15 comparing Remask RL and LLaDOU RL to support our convergence claims;
> > > 3. **W7 & Q4:** **Include plots of remask rate dynamics and reward variance** in Appendix C, Fig 13 and Fig. 15 across training to better illustrate RL stability; For RL training stablility, we only apply gradient clipping with a maximum norm of 1.0 which is described in Appendix B.2.4.
> > >
> > > **Q9:  Generalization to other DLMs: Since the model is based on LLaDA-8B, could this remasking mechanism generalize to other diffusion LMs such as Dream? If so, what architectural adjustments are needed?**
> > >
> > > The RemeDi remasking mechanism is **architecture-agnostic** and can be readily integrated into any existing diffusion LMs, including Dream. Here we provide a pipeline with minimal engineering overhead:
> > >
> > > (1) Implement the UPS to allow predicting per-token confidence $h$. It can be simply a few transformer/MLP layers on top of Dream's hidden states as an additional head.
> > >
> > > (2) Implement remask SFT and RL. For remask SFT, just perform additional token replacement when constructing inputs, and add a loss function $\mathcal{L}_\text{UPS}$. For RL, make one-line modification on LLaDOU so that the unmask policy selects all unmasked tokens from all positions, instead of only selecting the newly unmasked ones.
> > >
> > > You may directly adopt our code (https://github.com/iiiutch-ii/RemeDi) for other DLMs.
> > >
> > > **Q10: Random alternative tokens: In SFT, are random alternatives drawn uniformly from the vocabulary or sampled from the model’s top-k predictions?**
> > >
> > > We sample random alternative tokens uniformly from the vocabulary. While sampling from the model’s top-k predictions is a reasonable alternative, it would require an additional forward pass to compute token logits during SFT construction, which substantially increases training cost.
> > >
> > > **Q11: Inference efficiency: The paper states RemeDi achieves faster convergence; could you include inference latency comparisons (e.g., tokens per second vs. LLaDA), given that TPS + UPS may add computational overhead?**
> > >
> > > To directly address the comparison with LLaDA, we report their inference throughput under identical settings (bs=1, seqlen=1024, H800 GPU). RemeDi achieves 50.65 tok/s, more than 2× faster than LLaDA (24.15 tok/s), while also delivering substantially higher accuracy (GSM8K 89.1% vs. 78.3%). The speedup mainly arises from RemeDi's ability to perform variable-length block-wise generation with a block-causal attention mask. The more detailed throughput–performance curve is included in the Appendix C, Fig. 14.
> > >
> > > **Table V: Inference throughput comparison with LLaDA.**
> > >
> > > | Model            | Throughput (tokens/sec) | Speed up |
> > > | ---------------- | ----------------------- | -------- |
> > > | LLaDA            | 24.15                   | 1.00x     |
> > > | RemeDi           | 50.65                   | 2.10x    |
> > >
> > > ---
> > >
> > > References
> > >
> > > [1] Shao Z. et.al. Deepseekmath: Pushing the limits of mathematical reasoning in open language models. arXiv 2024
> > >
> > > [2] Touvron H et.al. Llama 2: Open foundation and fine-tuned chat models. arXiv 2023
> > >
> > > [3] Dubey A. et al. The llama 3 herd of models. arXiv 2024
> > >
> > > [4] Guo D. et.al. Deepseek-r1: Incentivizing reasoning capability in llms via reinforcement learning. arXiv 2025

---

### Official Review · Reviewer_Z76m · 2025-10-30

**Soundness:** 2
**Presentation:** 4
**Contribution:** 3
**Rating:** 4
**Confidence:** 4

**Summary:**

The paper investigates remasking as a way to improve masked diffusion language models. Remasking works by replacing some already-unmasked tokens with mask tokens, which subsequently get unmasked again. This way, uncertain or incorrect tokens can be revised and potentially fixed, thereby improving accuracy with minimal additional cost.

The proposed method, called RemeDi, introduces two new policies to identify the tokens to be remasked, one based on supervised fine-tuning (SFT) and one based on reinforcement learning (RL). This is done via a dual-stream transformer architecture: The token prediction stream (TPS) predicts unmasked tokens while the unmasking policy stream (UPS) predicts per-token confidence scores (for both masked and unmasked tokens). The resulting model achieves significant gains on a variety of benchmarks compared to baselines.

**Strengths:**

### S1. Strong benchmark performance
The proposed model achieves strong benchmark performance, beating many diffusion and autoregressive baselines of similar size.

### S2. Flexible post-training method
The proposed method is flexible enough to be generally applicable to many existing masked diffusion models, and can be combined with other optimizations such as block diffusion.

### S3. Analysis of remasking behavior
The analysis of the remasking behavior of the trained policies provides valuable insights. It sheds light into task difficulty as well as which part of the sampling process may be most prone to errors.

**Weaknesses:**

(ordered by decreasing severity)

### W1. Source of performance improvements is unclear
It is somewhat unclear which improvements stem from the continued (multi-task) pretraining and which come from the remasking during sampling. This question is partially addressed by comparing Remask SFT with vanilla SFT and Remask RL with LLaDOU RL, but some confounders remain. The multitask objective employed by Remask SFT may be beneficial even in a vanilla (or adaptive) inference setting where the model confidence is obtained heuristically. Similarly, the Remask RL approach is very similar to the one proposed by LLaDOU (Huang et al., 2025), so it’s unclear where exactly the improvements come from.
The former can easily be addressed by reporting the performance of Remask SFT using vanilla and adaptive masked diffusion sampling (Kim et al., 2025). The latter can be addressed by providing a detailed analysis of the differences between Remask RL and LLaDOU RL and, if necessary, ablating atomic changes to measure their individual impact (e.g. training vs. sampling improvements).

### W2. Reproducibility
As far as I can tell, there will be no model weights or codebase accompanying the paper, which is a major concern for reproducibility, especially given the clarity concerns regarding the source of the performance gains (W1).

### W3. Motivation and effect of dual-stream architecture
The motivation for using a dual-stream architecture over conceptually simpler approaches (e.g. a dedicated unmasking policy head) is unclear, and no ablations are performed on this. Similarly, the computational overhead of the proposed architecture (more parameters, slower forward pass) may provide an unfair advantage compared to baselines. The former can be addressed by providing appropriate ablations, whereas the latter calls for reporting inference speed of the proposed method compared to baselines in addition to benchmark accuracies.

### Conclusion
As presented, reasons to reject the paper outweigh reasons to accept: While the benchmark performance of the proposed model is impressive and beats state-of-the-art methods (S1), the lack of clarity regarding the source of those gains (W1) together with concerns about reproducibility (W2) and inference speed (W3) amount to significant concerns regarding soundness. I will be happy to increase my final score if these weaknesses can be addressed.

**Questions:**

- Q1. What is the effect of the multi-task objective (Eq. 5) together with incorrect token augmentation (Eq. 3)? More specifically, what is the performance of Remask SFT when using standard vanilla/adaptive sampling (Kim et al., 2025) compared to full Remask SFT? (also see W1)
- Q2. What is the difference between Remask RL and LLaDOU RL (Huang et al., 2025)? Specifically, is the sampling policy in Eq. 8 not identical to the one from Eq. 9 in LLaDOU? If so, where do the improvements of Remask RL over LLaDOU come from? (also see W1)
- Q3. Will the model weights and/or training code be open-sourced? (also see W2)
- Q4. How does the proposed dual-stream architecture compare to more naive baselines, e.g. a single unmasking policy head or a single additional Transformer block (as employed by LLaDOU)? (also see W3)
- Q5. What is the performance overhead associated with RemeDi compared to baselines? (also see W3)

Nits (not considered for final score):
- L31: Austin et al. (2022) should be cited for discrete diffusion.
- L34: Sahoo et al. (2024) and Shi et al. (2024) should be cited for masked diffusion.
- L103: typo: missing space after “Recent studies”
- L160: typo: “paradigm offers” -> “paradigms offer”
- L203: typo: extra space after “(Guo et al., 2025)”

---

### References
- Austin et al. (2022): https://arxiv.org/abs/2107.03006
- Huang et al. (2025): https://arxiv.org/abs/2505.10446
- Kim et al. (2025): https://arxiv.org/abs/2502.06768
- Sahoo et al. (2024): https://arxiv.org/abs/2406.07524
- Shi et al. (2024): https://arxiv.org/abs/2406.04329

---

> ### Author Response · Authors · 2025-12-03
>
> **Q1: What is the effect of the multi-task objective (Eq. 5) together with incorrect token augmentation (Eq. 3)? More specifically, what is the performance of Remask SFT when using standard vanilla/adaptive sampling (Kim et al., 2025) compared to full Remask SFT? (also see W1)**
>
> To isolate the effect of our multi-task objective and incorrect-token augmentation, we compared **vanilla sampler**, **adaptive sampler** and our **remask sampler** under the same Remask SFT model. The results are:
>
> **Table I: Effect of different samplers under the same Remask SFT model.**
>
> | Sampler  | GSM8K    | MATH-500 | HumanEval | MBPP     | IFEval   |
> | ------------- | -------- | -------- | --------- | -------- | -------- |
> | Vanilla | 86.3     | 38.3     | 43.9 | 55.4     | 69.2  |
> | Adaptive  | **86.6** | 40.3 | 43.9 | 55.4| 69.2 |
> | Remask (Ours) | 86.3     | **52.2** | **71.3**  | **57.8** | **81.9** |
>
> Vanilla and adaptive sampler yield almost identical behavior, showing limited benefit from the multi-task/augmentation signals. In contrast, our remask sampler shows higher performance—particularly on reasoning-intensive (MATH-500), code generation (HumanEval, MBPP) and instruct-following (IFEval).
>
> **Q2: What is the difference between Remask RL and LLaDOU RL (Huang et al., 2025)? Specifically, is the sampling policy in Eq. 8 not identical to the one from Eq. 9 in LLaDOU? If so, where do the improvements of Remask RL over LLaDOU come from? (also W1)**
>
> The core difference is that Remask RL defines the remask action over all tokens (allowing previously generated tokens to be remasked), whereas LLaDOU RL restricts the remask action to only currently masked positions (preserving all prior predictions). Consequently, RemeDi is able to **iteratively refine** its output by correcting earlier suboptimal token predictions, leading to higher performance.
>
> Here we compare them with identical notations. Considering that the model unmasks $K_n$ tokens (denoted the set of positions as $\mathcal{U}_n$) after step $n$,
>
> - LLaDOU RL samples **only** $\Delta K=K_n - K_{n-1}$ newly unmasked tokens $\Delta\mathcal{U}\_{n}=[u'\_n(1), ..., u'\_n(K)]$ from the set of masked positions $\mathcal{M}\_{n-1}$, keeping all previously $K\_{n-1}$ unmasked tokens fixed (i.e., $\mathcal{U}\_n=\mathcal{U}\_{n-1}\cup\Delta\mathcal{U}\_n$). Thus, it never remask previously generated tokens.
> $$
> \pi\_{\theta,n}^\{\text{unmask}}(\Delta\mathcal{U}\_n|x\_{n-1})=
> \prod\_{k=1}\^K\frac
> {\exp(h\_{\theta,n}\^{u'\_n(k)})}
> {\sum\_{j=k}\^K\exp(h\_{\theta,n}\^{u'\_n(j)}) +
> \sum\_{j\in \mathcal{M}\_n}\exp(h\_{\theta,n}\^{j})}
> $$
>
> - In Remask RL, by contrast, **all** $K_n$ unmask tokens $\mathcal{U}_n=[u_n(1), ..., u_n(K_n)]$ are sampled, meaning previously generated tokens may be remasked.
>
> $$
> \pi\_{\theta,n}\^\text{unmask}(\mathcal{U}\_n \mid x\_{t\_{n-1}})
>     = \prod\_{k=1}\^{K_n}
>       \frac{\exp(h\_{\theta,n}\^{u\_n(k)})}
>            {\sum\_{j \notin \{u\_n(1),\dots,u\_n(k-1)\}} \exp(h\_{\theta,n}\^j)} .
> $$
>
> **Q3: Will the model weights and/or training code be open-sourced? (also W2)**
>
> We have provided an anonymous repository link containing both the code and pretrained model in the paper (https://github.com/iiiutch-ii/RemeDi.).
>
> **Q4. How does the proposed dual-stream architecture compare to more naive baselines, e.g. a single unmasking policy head or a single additional Transformer block (as employed by LLaDOU)? (also W3)**
>
> We conduct an additional comparison by adding only one transformer block to predict $h$ after the final layer of TPS. As shown in Table below, following the same ablation study settings as in Appendix B.2.5,  the dual-stream architecture employed by RemeDi consistently achieved higher performance across different benchmarks.
>
> **Table II: Dual-stream UPS vs. a single additional Transformer block.**
>
> | UPS architecture | GSM8K    | MATH-500 | HumanEval | MBPP |
> | --------------------- | -------- | -------- | --------- | -------- |
> | One transformer block | 78.6     | 40.7     | 47.0      | 43.8   |
> | Dual-stream  | **83.6** | **42.7** | **50.0**  | **44.0** |
>
> **Q5: What is the performance overhead associated with RemeDi compared to baselines? (also W3)**
>
> We acknowledge that adding the UPS module introduces additional overhead compared to the baseline **RemeDi (w/o UPS)**. However, when anchored to **LLaDA**, the slowdown is modest: RemeDi with UPS remains **over 2× faster** than LLaDA under identical settings (batch size = 1, sequence length=1024 on a single H800 GPU). The higher throughput mainly arises from
> RemeDi's ability to perform variable-length block-wise generation with a block-causal attention mask. The throughput comparison is shown in Table III.
>
> **Table III: Throughput comparison with and without the UPS component.**
>
> | Model  | Throughput (tokens/sec) | Speed up |
> | ---------------- | ----------------------- | -------- |
> | LLaDA    | 24.15  | 1.00x     |
> | RemeDi    | 50.65  | 2.10x    |
> | RemeDi (w/o UPS) | 63.26| 2.62x    |

---

### Official Review · Reviewer_arzr · 2025-10-31

**Soundness:** 3
**Presentation:** 3
**Contribution:** 3
**Rating:** 6
**Confidence:** 4

**Summary:**

RemeDi augments diffusion LMs with a learnable remasking policy: a per-token confidence head (UPS) decides which tokens to unmask or re-mask, trained via remask-SFT then trajectory-level RL, yielding consistent gains on math/code/general tasks under block-wise variable-length decoding.

**Strengths:**

1. Turns remasking from an inference hack into a learned policy; per-token confidence (UPS) guides when to unmask or re-mask.
2. Two-stage training (remask-SFT + RL) aligns token-level corrections with sequence-level rewards; compatible with monotonic denoising.
3. Works atop block-wise variable-length decoding (LLaDA-style); integrates without changing the base noise family.
4. Confidence head provides interpretable signals and a knob to trade exploration vs. commitment.
5. Orthogonal to many decoding/acceleration tricks; likely to stack with other DLM advances.

**Weaknesses:**

1. No clear accounting of UPS params/FLOPs/memory or train/infer throughput; lacks equal-quality step/latency comparisons to AR/other DLMs.
2. Missing UPS structure/attachment ablations (bi-residuals, zero-init bridge, layer choice) and sensitivity to ρ_(t,"incorrect" )  and ratio r.
3. RL uses preference and verifiable rewards; robustness and transfer to new domains remain uncertain.
4. Sparse head-to-head with edit-flow/seed-diffusion/predictor-corrector under identical settings.
5. Potential remask oscillation (low-confidence flip-flops); termination/annealing not systematically reported.
6. Higher engineering complexity (dual streams + two-stage training) raises adoption barriers.

**Questions:**

1. Report UPS overhead (params/FLOPs/memory) and equal-quality latency/tokens-per-second vs. AR/strong DLM baselines, with quality–latency Pareto curves on fixed hardware.
2. Provide ablations for UPS attachments and components (bi-residuals, zero-init bridge, layer choice) plus sensitivity sweeps of (\rho_{t,\text{incorrect}}) and ratio (r).
3. Decompose reward contributions, test cross-domain transfer without retuning, and include robustness/calibration (e.g., ECE) under distribution shift.
4. Add unified head-to-head comparisons with edit-flow, seed-diffusion, and predictor-corrector using identical checkpoints, datasets, and decoding budgets.
5. Quantify remask oscillation and compare termination/annealing (fixed/decayed thresholds, hysteresis) to show stability–quality trade-offs.
6. Summarize engineering cost (LoC, stages, time, hardware) and release a minimal plug-and-play UPS recipe with an inference-only lightweight variant.

---

> ### Author Response · Authors · 2025-12-03
>
> **W1 & Q1: No clear accounting of UPS params/FLOPs/memory or train/infer throughput; lacks equal-quality step/latency comparisons to AR/other DLMs. / Report UPS overhead (params/FLOPs/memory) and equal-quality latency/tokens-per-second vs. AR/strong DLM baselines, with quality–latency Pareto curves on fixed hardware.**
>
> UPS adds **939.58M parameters**  upon LLaDA 8B (approximately +1.88 GB memory when loaded in BF16). The UPS module consists of four Transformer blocks, so its additional FLOPs are roughly **1/8 of the base model** with 32 blocks, making its computational cost minor relative to end-to-end inference.
>
> To quantify runtime cost, Fig. 14 in Appendix C compares both throughput and accuracy on GSM8K against AR and DLM baselines. By varying the number of tokens denoised per step, RemeDi supports a flexible quality–latency trade-off. With the one-token-per-step setting, it reaches **50.65 tok/s** and **89.1%** on GSM8K. When decoding 2 tokens per step, throughput increases to **99.82 tok/s** with only a small accuracy drop to **88.3%**. As shown in Fig. 14, RemeDi forms a better Pareto frontier than both strong AR and DLM baselines.
>
> All throughput measurements use **batch size = 1, sequence length = 1024** on a single **H800 GPU**, ensuring a fair and consistent comparison.
>
> **W2 & Q2: Missing UPS structure/attachment ablations (bi-residuals, zero-init bridge, layer choice) and sensitivity to $\rho_{t,\text{incorrect}}$ and ratio $r$. / Provide ablations for UPS attachments and components (bi-residuals, zero-init bridge, layer choice) plus sensitivity sweeps of $\rho_{t,\text{incorrect}}$ and ratio $r$.**
>
> We conduct ablation studies on key UPS components under the same training setup described in Appendix B.2.5. As shown in Table I, Either removing bi-residual connections or the initializing their weights other than zeros leads to performance drops, especially on GSM8K and HumanEval, demonstrating the necessity of both components.
>
> **Table I: Ablation studies on the UPS structure**
>
> |                      | GSM8K    | MATH-500 | HumanEval | MBPP     |
> | -------------------- | -------- | -------- | --------- | -------- |
> | RemeDi (+Remask SFT) | **83.6** | 42.7     | **50.0**  | 44.0     |
> | w/o bi-residuals     | 76.6     | **43.1** | 45.7      | 42.2     |
> | w/o zero-init        | 75.2     | 42.7     | 45.1      | **44.8** |
>
> As to the layer choice, we compare our multi-layer feature aggregation with a simpler alternative that directly feeds only the final TPS hidden states into UPS.
> At shown in Table II, despite having comparable parameter counts, multi-layer aggregation yields consistently better performance across different benchmarks.
>
> **Table II: Ablation studies on the layer choice**
>
> | Layer Choice          | GSM8K    | MATH-500 | HumanEval | MBPP     |
> | --------------------- | -------- | -------- | --------- | -------- |
> | After Final Layer     | 78.6     | 40.7     | 47.0      | 43.8     |
> | Multi-Layer Aggregation    | **83.6** | **42.7** | **50.0**  | **44.0** |
>
> For the noise schedule to replace tokens when constructing noisy inputs, as introduced in Section 3.2.1, our algorithm only requires that the number of incorrect tokens is strictly smaller than the number of mask tokens:
> $$\lceil \rho_{t,\text{incorrect}}\cdot  (1 - \rho_{t,\text{mask}})\cdot L \rceil <
> \lceil \rho_{t,\text{mask}}\cdot L \rceil,$$
> which ensures that the noise level decreases monotonically during the reverse diffusion process. Here we directly adopt $\rho_{t,\text{incorrect}}=4r\cdot t(1-t)$ amd $r=0.1$ from GIDD [1] for a fair comparison, which satisfies the requirement above. We do not manually tune these hyperparameters. Despite this, Remask SFT already demonstrates its superiority over other SFT algorithms, indicating that this method is robust and not sensitive to the particular choice of $\rho_{t,\text{incorrect}}$ and $r$.
>
>
> **W3 & Q3:  RL uses preference and verifiable rewards; robustness and transfer to new domains remain uncertain. / Decompose reward contributions, test cross-domain transfer without retuning, and include robustness/calibration (e.g., ECE) under distribution shift.**
>
> To evaluate the robustness of the model after Remask RL and its ability to generalize to new domains, we assess the model's performance both before and after Remask RL on a new benchmarks, Arena-Hard 2.0 [2]. It contains 500 fresh, challenging and multilingual real-world user queries. Notably, our training data did not include multilingual data. Even so, when taking GPT-O3-mini as the baseline model, the win rate of RemeDi increases from 0.3 to 0.6 after training with Remask RL, using GPT-4.1 as the judge model. This also represents a a relative performance improvement exceeding 50% over the original LLaDA (0.4).

---

> ### Author Response · Authors · 2025-12-03
>
> **W4 & Q4: Sparse head-to-head with edit-flow/seed-diffusion/predictor-corrector under identical settings. / Add unified head-to-head comparisons with edit-flow, seed-diffusion, and predictor-corrector using identical checkpoints, datasets, and decoding budgets.**
>
> Here we provide two separate sets of comparisons.
>
> **Training with Seed Diffusion.**  Since the official Seed Diffusion model and implementation are not publicly available, we do our best to reproduce it. For a fair comparison, we train seed diffusion under the same base model and identical training configuration as in Appendix B.2.5. Our reproduced implementation is also provided together with our codebase (https://github.com/iiiutch-ii/RemeDi). As shown in Table III, Remask SFT consistently outperforms Seed Diffusion, demonstrating the advantage of learning an explicit remasking policy during training.
>
> **Table III: Unified head-to-head comparison with other training algorithms under identical settings.**
>
> | Method  | GSM8K    | MATH-500 | HumanEval | MBPP     |
> | -------------- | -------- | -------- | --------- | -------- |
> | Baseline       | 80.3     | 34.7     | 41.5      | 42.6     |
> | Vanilla SFT    | 83.1     | 40.1     | 48.2      | 43.4     |
> | Seed Diffusion | 63.9     | 28.0     | 5.4       | 9.8      |
> | Remask SFT     | **83.6** | **42.7** | **50.0**  | **44.0** |
>
> **Inference with predictor-corrector samplers.** We apply a representative predictor-corrector sampler, ReMDM [3], to RemeDi-Instruct. The evaluation setup is the same as in Appendix B.3.1. The results in Table IV show that our learned remasking policy (via Remask SFT) is more effective than the random remasking strategy employed in predictor-corrector samplers.
>
> **Table IV: Comparison between our learned remask policy in RemeDi and the ReMDM predictor-corrector, both evaluated with RemeDi-Instruct.**
>
> | Method  | GSM8K | MATH-500 | HumanEval | MBPP |
> | -------------- | -------- | -------- | --------- | -------- |
> | RemeDi + predictor-corrector | 58.3  | 38.7  | 39.6  | 54.2|
> | RemaDi (Ours) | **86.3** | **52.2** | **71.3**  | **57.8** |

---

> ### Author Response · Authors · 2025-12-03
>
> **W5 & Q5: Potential remask oscillation (low-confidence flip-flops); termination/annealing not systematically reported. / Quantify remask oscillation and compare termination/annealing (fixed/decayed thresholds, hysteresis) to show stability–quality trade-offs.**
>
> To address the reviewer’s concern about potential remask oscillation, we provide a quantitative analysis in Appendix C, Fig. 12. We report the **number of times each token is remasked** during the diffusion process. Approximately **24.5%** of tokens are remasked more than once, but only **12%** are remasked more than twice, and **95.5%** of all tokens are remasked no more than three times. This indicates that oscillatory flip-flop behavior is rare and remains well-controlled.
>
> We further analyze the relationship between the **remask ratio** and **timestep** within each block. We compute the average remask ratio across all blocks at each diffusion step on GSM8K. As shown in Appendix C, Fig. 13, the remask ratio initially increases in early steps and then gradually decreases as the diffusion process converges. Since our method explicitly defines the number of unmasked tokens $K_n$ at diffusion step $n$, and $K_n$ increases linearly from $0$ to $L$. Thus, if a previously unmasked token is remasked due to low confidence, another higher-confidence token must be unmasked to maintain the unmasked tokens number $K_n$. Consequently, the termination behavior of remasking is stable and does not exhibit the problematic oscillation patterns the reviewer is concerned about.
>
> **W6 & Q6: Higher engineering complexity (dual streams + two-stage training) raises adoption barriers. / Summarize engineering cost (LoC, stages, time, hardware) and release a minimal plug-and-play UPS recipe with an inference-only lightweight variant.**
>
> **Engineering cost**: Integrating the UPS module requires fewer than 50 lines of code (LoC) in modifications.  SFT loss code requires $<100$ LoC, and remask RL sample and loss code requires $<50$ LoC modifications built upon the prior work LLaDOU. The computational costs are approximately 672 and 32 H800-days for SFT and RL, respectively. We released our training code in https://github.com/iiiutch-ii/RemeDi.
>
> Given a DLM, the minimal pipeline to follow this paper is:
>
> (1) Implement the UPS to allow predicting per-token confidence $h$. We can simply add a few transformer/MLP layers on top of the TPS hidden states as an additional head.
>
> (2) Modify code for remask SFT and RL. For remask SFT, just perform additional token replacement and add a loss function $\mathcal{L}_\text{UPS}$. For RL, make one-line modification on LLaDOU so that the unmask policy select all unmasked tokens from all positions, instead of only selecting the newly unmasked ones.
>
>
> ---
>
> References
>
> [1] von Rütte D. et.al. Generalized interpolating discrete diffusion. arXiv 2025
>
> [2] Li, T. et. al. From Crowdsourced Data to High-Quality Benchmarks: Arena-Hard and BenchBuilder Pipeline. arXiv 2024
>
> [3] Wang, G.et. al. Remasking discrete diffusion models with inference-time scaling. arXiv 2025

---

### Official Review · Reviewer_4oJS · 2025-11-03

**Soundness:** 3
**Presentation:** 2
**Contribution:** 3
**Rating:** 4
**Confidence:** 3

**Summary:**

The paper diagnoses a structural limitation of mask-based diffusion language models (DLMs)—namely that once a token is unmasked it becomes effectively immutable, which prevents the model from performing self-reflection and post-hoc correction of its outputs. To mitigate this, the authors propose a self-reflective remasking mechanism (RemeDi) that augments the diffusion process with learnable per-token confidence scores and an explicit remasking operation that can reintroduce previously unmasked tokens for further refinement. They train the mechanism with a two-stage procedure—supervised Remask SFT followed by policy-style Remask RL—so the model learns when and how to re-mask and revise. Crucially, RemeDi preserves the diffusion model’s noise-monotonicity (the monotone decrease of corruption during denoising) while enabling iterative edit-and-refine behavior, and the paper reports substantial empirical improvements on mathematical reasoning, code generation, and general language tasks.

**Strengths:**

1. RemeDi enables self-reflective iterative refinement by jointly predicting per-token confidence scores and remasking low-confidence tokens so that previously revealed tokens can be selectively re-sampled without breaking the diffusion noise schedule.

2. The paper introduces a principled two-stage training pipeline, with supervised Remask SFT to teach detection and remasking and outcome-based Remask RL to optimize whole-generation trajectories, which yields stronger performance.

3. The remasking mechanism supports a wide range of edit behaviors such as replacement, insertion, deletion, merging and splitting, and produces empirical gains on mathematics, code generation, and general instruction benchmarks compared to prior open-source diffusion language models.

**Weaknesses:**

1. No reproduction code is provided; please supply an anonymous repository link or include the code in the supplementary materials.
2. The model diagram in the appendix is unclear — please clarify how $p$ and $h$ are predicted simultaneously and whether any new modules were introduced.
3. Please add a baseline that performs RL directly on LLaDA using the same datasets you used.

**Questions:**

1. I suggest moving some experimental results to the appendix and placing the main architecture figure in the main text.
2. Why did you run experiments with LLaDA rather than LLaDA 1.5?
3. Is inference latency substantially slower compared to the baseline?

---

> ### Author Response · Authors · 2025-12-03
>
> **W1: No reproduction code is provided; please supply an anonymous repository link or include the code in the supplementary materials.**
>
> We have provided an anonymous repository link containing both the code and model in the paper ([https://github.com/iiiutch-ii/RemeDi](https://github.com/iiiutch-ii/RemeDi)).
>
> **W2 & Q1: The model diagram in the appendix is unclear — please clarify how $p$ and $h$ are predicted simultaneously and whether any new modules were introduced. / I suggest moving some experimental results to the appendix and placing the main architecture figure in the main text.**
>
> During inference, UPS blocks are periodically inserted to produce an auxiliary representation $f_\text{UPS}$ for confidence scoring. The two streams exchange features: Each block in UPS receives inputs augmented with the hidden state in TPS $f_\text{TPS}$, and its output is also added back to $f_\text{TPS}$ after passing through a linear projection. Finally, $p$ and $h$ are predicted simultaneously via two separate linear heads applied to $f_\text{TPS}$ and $f_\text{UPS}$, respectively.
>
> We have modified the main architecture figure to make it more clear, and moved it in the main text. As shown in Figure 2, all modules in the TPS column are inherited from LLaDA, while other modules in UPS are newly introduced.
>
> **W3:  Please add a baseline that performs RL directly on LLaDA using the same datasets you used.**
>
> We have conducted an additional experiment applying reinforcement learning (RL) on LLaDA using the exact same training datasets and a compute budget of 32 H800 days as in our Remask RL stage. The results are summarized below:
>
> **Table I: Comparison of LLaDA + DCoLT vs. RemeDi (same data and compute cost)**.
>
> |                       | GSM8K     | MATH      | HumanEval | MBPP      | IFEval    |
> | --------------------- | --------- | --------- | --------- | --------- | --------- |
> | LLaDA + DCoLT         | 83.5%     | 36.8%     | 43.9%     | 41.2%     | 72.5%     |
> | RemeDi (++ Remask RL) | **89.1%** | **52.9%** | **73.2%** | **59.4%** | **85.4%** |
>
> **Q2: Why did you run experiments with LLaDA rather than LLaDA 1.5?**
>
> We conduct experiments on LLaDA for a fair comparison with other works, such as d1, wd1 and LLaDOU, which are also trained from LLaDA.
> Moreover, our training algorithm can also be applied on LLaDA 1.5 to further improve its performance. For example, after training LLaDA 1.5 using Remask RL on GSM8K with the following settings — 64 denoising steps, a generation length of 256 and a block length of 64. The accuracy is further improved to **83.94%**, which is **15.38%** higher than its original performance (68.56%).
>
> **Q3: Is inference latency substantially slower compared to the baseline?**
>
> We measure the throughput of different models with batch size = 1, sequence length = 1024 on a single H800 GPU. The results are as follows:
>
> **Table II: Inference throughput comparison.**
>
> | Model  | Throughput (tokens/sec) | Speed up |
> | ------ | ----------------------- | -------- |
> | LLaDA  | 24.15                   | 1.0x     |
> | RemeDi | 50.65                   | 2.1x    |
>
> As shown, the RemeDi achieves a high inference speed—over 2× faster than the original LLaDA. The higher throughput mainly arises from RemeDi's ability to perform variable-length block-wise generation with a block-causal attention mask.

---

### Meta-Review · Area_Chair_5yXn · 2026-01-07

**Summary:**

RemeDi introduces a self-reflective remasking mechanism for diffusion language models using a dual-stream architecture (TPS for token prediction, UPS for confidence scoring) that enables iterative refinement of generated tokens. Training follows a standard two-stage SFT→RL pipeline. The work achieves state-of-the-art results among open-source DLMs across math, code, and reasoning benchmarks.

Initial scores were 4/6/4/4. Authors provided comprehensive responses addressing major concerns, including code/model release, new experiments, and detailed ablations. No reviewers engaged during discussion, but the technical merit of responses is clear.

**Reviewer Concerns:**

Concerns Successfully Addressed:
- Full code and model release to ensure reproducibility. (Reviewers 4oJS, Z76m, AeNa)
- Comprehensive ablations, including the UPS structure and additional head-to-head comparisons (Seed Diffusion, ReMDM). (Reviewers arzr, Z76m, AeNa)
- Source of improvements, isolating the remasking mechanism effectiveness via sampler comparison on identical model. (Reviewer Z76m)
- Inference-time analysis, reporting latency and quality–throughput Pareto frontiers. (Reviewers 4oJS, arzr, AeNa)
- Paper revisions, improving architecture diagram and adding reward curve, remask-rate dynamics, and throughput plot. (Reviewers 4oJS, arzr, AeNa)

Outstanding Concerns:
I think the rebuttal has well addressed all major concerns.

**Reviewer Scores:**

- Reviewer 4oJS: 6/10 (updated from 4).
- Reviewer arzr: 6-8/10 (updated from 6).
- Reviewer Z76m: 6/10 (updated from 4).
- Reviewer AeNa: 6/10 (updated from 4).

---

### Decision · Program_Chairs · 2026-01-26

Accept (Poster)